

# Updated global and regional trends of stratospheric ozone profiles

Viktoria F. Sofieva[1], Monika E. Szelag[1], Natalya Kramarova[2], Robert Damadeo[3], Wolfgang Steinbrecht[4], Irina Petropavlovskikh[5,18], Corinne Vigouroux[6], Eliane Maillard Barras[7], Daniel Zawada[8], Kleareti Tourpali[9], Stacey M. Frith[2,10], Jeannette D. Wild[11, 12], Sean M. Davis[13], Carlo Arosio[14], Mark Weber[14], Alexei Rozanov[14], Brian Auffarth[14], Lucien Froidevaux[15], Ryan Fuller[15], Doug Degenstein[8], Kimberlee Dube[8], Peter Effertz[5,18], Thierry Leblanc[16], Gérard Ancellet[17], Sophie Godin-Beekmann[17] , Glen McConville[5,18], Richard Querel[19], Dan Smale[19], Marie-Renee DeBacker[20], Emmanuel Mahieu[21], Ralf Sussmann[22]

[1] Finnish Meteorological Institute, Helsinki, Finland

[2] NASA Goddard Space Flight Center, Greenbelt, MD, USA

[3] NASA Langley Research Center, Hampton, VA, USA

[4] DWD (German Weather Service), Hohenpeissenberg, Germany

[5] Cooperative Institute for Research in Environmental Science (CIRES), University of Colorado-Boulder, Boulder, CO, USA

[6] Royal Belgian Institute for Space Aeronomy (BIRA-IASB), Uccle, Belgium

[7] Federal Office of Meteorology and Climatology, MeteoSwiss, Payerne, Switzerland

[8] Institute of Space and Atmospheric Studies, University of Saskatchewan, Saskatoon, Canada

[9] Aristotle University, Thessaloniki, Greece

[10] Science Systems and Applications, Inc, Greenbelt, MD, USA

[11] Earth System Science Interdisciplinary Center (ESSIC/CISESS), University of Maryland, College Park, MD, USA

[12] NOAA/NESDIS/Center for Satellite Applications and Research (STAR), College Park, MD, USA

[13] NOAA Chemical Sciences Laboratory, Boulder, CO, USA

[14] Institute of Environmental Physics, University of Bremen, Germany

[15] Jet Propulsion Laboratory, California Institute of Technology, Pasadena, California, USA

[16] Jet Propulsion Laboratory, California Institute of Technology, Wrightwood, CA, USA

[17] Laboratoire Atmosphères Observations Spatiales (LATMOS), CNRS, Sorbonne Université, Paris, France

[18] NOAA Global Monitoring Laboratory, Boulder, CO, USA

[19] New Zealand Institute for Earth Science Limited, Lauder, New Zealand

[20] Université de Reims Champagne-Ardenne, France

[21] Department of Astrophysics, Geophysics and Oceanography, UR SPHERES, University of Liège, Liège, Belgium

[22] Karlsruhe Institute of Technology (KIT), IMK-IFU, Garmisch-Partenkirchen, Germany

*Correspondence to*: Viktoria F. Sofieva (viktoria.sofieva@fmi.fi)



**Abstract.** We present updated evaluation of stratospheric ozone profile trends in the 60°S-60°N latitude range using long-term ground-based and satellite climate data records, as well as simulations by chemistry-climate models. The trends are evaluated using the LOTUS (Long-term Ozone Trends and Uncertainties in the Stratosphere) regression model.

Analyses of satellite data confirm the statistically significant positive ozone trends in the period 2000-2024 in the upper stratosphere of ~1-3 %decade$^{-1}$, with larger trends at mid-latitudes compared to the tropics. The trends are slightly positive or close to zero in the middle stratosphere, and mostly negative, -1-2 %decade$^{-1}$, in the lower stratosphere; but they are not statistically significant. The morphology and magnitude of ozone trends are similar to previous analyses (2000-2020 trends).

Ozone trends in 2000-2024 predicted by climate model simulations are in good agreement with combined satellite trends. In the upper stratosphere, models predict a slightly stronger ozone recovery than observations. In the lower stratosphere, both models and satellite observations report negative trends of in the tropics, while modelled ozone trends are slightly positive at mid-latitudes.

Ozone profile trends over several stations estimated from ground-based records capture the same overall vertical pattern of ozone trends as merged gridded satellite datasets.

Analyses of regional ozone profile trends in 2003-2024 using merged satellite datasets confirmed the previous observations of a longitudinal structure in ozone trends in the NH mid-latitude stratosphere, with positive trends over Scandinavia and negative trends over Siberia. However, the magnitude of this dipole-like structure is reduced compared to previous analyses.

# 1    Introduction

The ozone layer protects life on Earth from harmful ultraviolet solar radiation and plays an important role in the radiation budget of the atmosphere (Brasseur and Solomon, 2005; WMO, 2022). In the late 20th century, human emissions of ozone-depleting substances (ODS) adversely affected the ozone abundance in the atmosphere, most notably resulting in the annually recurring Antarctic ozone hole. The Montreal Protocol, signed in 1987, and its Amendments have curbed the amount of ODSs in the atmosphere, resulting in the slow recovery of the ozone layer( Godin-Beekmann et al., 2022; Solomon et al., 2016; Steinbrecht et al., 2017; WMO, 2022 and references therein). Despite these global efforts, the recovery of the ozone layer remains uneven and vulnerable to emerging threats such as climate change and unregulated emissions, therefore the need for continuous monitoring of the ozone layer should be recognized. The stratospheric ozone abundance and its vertical distribution is regularly measured using ground-based, in-situ and satellite instruments.

The quadrennial World Meteorological Organization (WMO) Ozone Assessments (since 1985) provide comprehensive scientific reports on the state of the ozone layer and its recovery progress. According to the latest Assessment (referred to as WMO-2022 hereafter, (WMO, 2022)) recovery of ozone in the upper stratosphere is progressing.

Ozone profile measurements allow for the evaluation of trends in the vertical distribution of ozone, which is important for understanding the processes governing the evolution and variability of the ozone abundance. According to WMO-2022



and Godin-Beekmann et al. (2022) (hereafter referred to as GB22), ozone is increasing in the upper stratosphere (at 2-3 hPa) at a rate of ~ 2 % decade$^{-1}$, and these trends are significant at the 95% confidence level. These changes are in good agreement with predictions by the chemistry-transport models. Statistically insignificant negative ozone trends are derived in the lowermost stratosphere, with the most pronounced negative trends in the tropics. While negative tropical upper troposphere

and the lower stratosphere (UTLS) ozone trends are expected due to model-predicted acceleration of the Brewer-Dobson circulation, the observed negative trends in the mid-latitude lower stratosphere disagree with mostly positive trends predicted by the models (Ball et al., 2018, 2020, GB22, WMO-2022). However, trend uncertainties are large below ~ 20 km, and all trend estimates are not statistically significant.

The LOTUS (Long-term Ozone Trends and Uncertainties in the Stratosphere) activity under APARC (Atmospheric

Processes and their Role in Climate), since 2016, aims to improve our understanding of stratospheric ozone recovery by updating observations, evaluating uncertainties in trend analyses from different datasets, and providing the information for the WMO ozone assessments. Within this activity, an open-source regression model for evaluation of ozone trends has been created (https://usask-arg.github.io/lotus-regression/). It has been used for evaluation of ozone profile trends for the WMO assessments since 2018 (WMO, 2018, 2022).

Our paper is a follow-up to previous studies (Petropavlovskikh et al., 2019; referred hereafter as P19) and GB22. Its main objective is to provide detailed and updated information about the vertical distribution of ozone trends in the stratosphere. Our analyses are based on long-term climate data records of ozone profiles from ground-based and merged satellite data. Since WMO-2022, these climate data records have been updated and improved by using the latest versions of the data from individual instruments, and new datasets have become available. In this paper, we evaluate zonally averaged trends of ozone profiles and

compare them with the results of the WMO-2022 assessment. In addition to the standard presentation of ozone trends in % decade$^{-1}$, we also present ozone trends in absolute units of DU km$^{-1}$ decade$^{-1}$, which allows visualization of contribution of atmospheric layers to the total ozone column trends. The trends from observations are compared with trends predicted by the climate models. We also analysed regional trends of ozone profiles, similarly to past work (Arosio et al., 2019; Sofieva et al., 2021; WMO, 2022) but with updated and new datasets.

The paper is organized as follows. Observation data sets and model simulations are described in Section 2. Section 3 is dedicated to the updates in the LOTUS regression model and evaluation of trends. The obtained trends are discussed in Section 4. Section 5 provides a summary to conclude the paper.

## 2 Observations and models

### 2.1 Merged satellite datasets

The basic information on merged satellite datasets used in our paper is collected in Table 1. In addition to the zonal mean merged satellite datasets used in GB22 and WMO-2022 - SBUV MOD, SBUV COH, GOZCARDS, SWOOSH, SAGE-CCI-





OMPS, SAGE-OSIRIS-OMPS, SAGE-SCIAMACHY-OMPS – we also included a new SAGEII-OSIRIS-SAGEIII dataset (Bognar et al., 2022).

Compared to WMO-2022 (Table 1), the majority of the merged datasets used updated versions of individual satellite ozone

records. The SAGE-CCI-OMPS+ dataset (Sofieva et al., 2023) has been updated by including data from two additional satellite instruments - POAM III and SAGE III/ISS, and OMPS-LP profiles processed by the University of Bremen. The updated version of SBUV-COH dataset includes OMPS data from both Suomi-NPP and NOAA-20 satellites. In GOZCARDS, Aura MLS v5 data are used starting Jan. 1, 2024, as opposed to v4 data prior to this. In fact, the ozone profiles from MLS v5 have negligible zonal mean offsets in comparison to MLS v4 (but Aura MLS v4 data were no longer produced after May 31, 2024).

A relevant update affecting several merged limb satellite data records was the release of a new version of the Level 1 data (gridded radiances) for the Suomi NPP OMPS Limb Profiler in 2023 (Jaross, 2023). Version 2.6 includes a critical correction to the instrument altitude registration drift that was found in version 2.5. Kramarova et al. (2024) reported that the analysis of the version 2.6 ozone record demonstrated the improved stability of the LP ozone data, as confirmed by the reduced relative drifts between the LP ozone and correlative measurements. The same conclusion on improved stability of ozone profiles is

done in Arosio et al. (2024).

**Table 1. Information about the merged satellite datasets used in the paper.**

| Dataset and references | Coverage and spatio-temporal resolution | Composition | Updates after WMO-2022 |
|---|---|---|---|
| **SBUV-MOD v8.7** (Frith et al., 2014) | 1970 – present 5° lat x 1 month 50 – 0.5 hPa | BUV v8.7 on Nimbus-4, SBUV v8.7 on Nimbus-7 & SBUV/2 v8.7 on NOAA-11, 14, 16, 17, 18, 19; NASA OMPS NP v2.9 on S-NPP. | Updated version of OMPS NP |
| **SBUV-COH v8.6** | 1979-present 5° lat x 1 month 50 – 0.5 hPa | SBUV v8.6 on Nimbus-7 & SBUV/2 v8.6 on NOAA-9, -11, 16, 17, 18, 19; OMPS v4r5 on S-NPP and NOAA-20 | Updated version of OMPS (was v4r1); Added NOAA-20 data from 2020 onward. |
| **GOZCARDS v2.2** (Froidevaux et al., 2015) | 1979-present 10° lat x 1 month 215 – 0.2 hPa | SAGE I v5.9_rev, SAGE II v7, HALOE v19, Aura MLS v4 and v5 (for 2024 onward) | Starting Jan. 1, 2024, the Aura MLS v5 data are used. |
| **SWOOSH v2.71** (Davis et al., 2016) | 1984-present 10° lat x 1 month 316 – 1 hPa | SAGE II v7, UARS MLS, UARS HALOE, Aura MLS v5 | Updated version of MLS |



| SAGE-CCI-OMPS+ (Sofieva et al., 2017, 2023) | 1984-present, 10° lat x 1 month 10 – 50 km | SAGE II v7, POAM III v4, OSIRIS v7.4, MIPAS KIT v8, GOMOS ALGOM2s v 1.0, SCIAMACHY v3.5, ACE-FTS v5.2, OMPS USask v 1.3.0 and UBr v4.1, SAGE III/ISS v 5.3 | updated versions of OSIRIS, OMPS-LP, ACE-FTS, SAGE III/ISS, added data from POAM III and SAGE III/ISS, OMPS UBr |
|---|---|---|---|
| SAGE-OSIRIS-OMPS (Bourassa et al., 2018) | 1984-present 10° lat x 1 month 10 – 50 km | SAGE II v7, OSIRIS v7.4, OMPS USask v1.3.0 | updated versions of OSIRIS and OMPS-LP |
| SAGEII-OSIRIS-SAGEIII (Bognar et al., 2022) | 1984-present 10° lat x 1 month 10 – 50 km | SAGE II v7, OSIRIS v7.4, SAGE III/ISS v6, sampling bias correction | New dataset |
| SAGE-SCIAMACHY-OMPS (Arosio et al., 2019) | 1984-present 10° lat x 1 month 8.5 – 60.5 km | SAGE II v7, SCIAMACHY v3.5, OMPS UBr v4.1 | updated version of OMPS-LP |
| MEGRIDOP (Sofieva et al., 2021) | 2001-present 10° lat x 20° lon x 1 month 10 – 50 km | OSIRIS v7.4, MIPAS KIT v8, GOMOS ALGOM2s v 1.0, SCIAMACHY v3.5, OMPS USask v 1.3.0, MLS v5 | updated versions of OSIRIS, OMPS-LP, MLS |
| SCIAMACHY-OMPS (Arosio et al., 2019) | 2002-present 5° lat x 20° lon x 1 month 8.5 – 60.5 km | SCIAMACHY v3.5, OMPS UBr v4.1 | updated version of OMPS-LP |

## 2.2    Ground-based ozone climate data records

Ground-based data records used in this study are similar to those used in GB22 and have been extended to 2024 (see Table 2). The lidar ozone record at Mauna Loa stopped in 2022 due to the eruption of Mauna Loa that damaged the access to the observatory.

Dobson Umkehr data are derived from the zenith sky observations using the optimal estimation technique (Petropavlovskikh et al., 2005, 2022). NOAA historical (up to 2020) Umkehr Dobson data were homogenized at 5 stations to remove instrumental

artifacts and biases (Petropavlovskikh et al., 2022). The Arosa/Davos MeteoSwiss data record has been homogenized using colocated Brewer Umkehr records and is described in Maillard Barras et al. (2022). For this study all Umkher data records



were extended through 2024. The trends for the 2000-2020 period were analysed using the LOTUS MLR model and were found to generally to agree (within uncertainty) with ozonesonde and COH overpass trends (Petropavlovskikh et al., 2025). The comparisons of the MLR and DLM trends over Arosa/Davos are summarized in Maillard Barras et al. (2022).

The three time-series at the FTIR stations (Jungfraujoch, Zugspitze, and Lauder) have been entirely reprocessed using an updated retrieval strategy as described in Björklund et al. (2024). In this study focusing on intercomparisons of many instruments measuring ozone at Lauder, it was shown that the trends using the updated and reprocessed FTIR updated data were more consistent with other measurements than previously reported, e.g., in GB22. The Mauna Loa microwave radiometer data record was interrupted in 2022 and resumed in January 2025 and is nevertheless used in this study. Since GB22, the Bern

microwave radiometer data record has been reprocessed for harmonization with the Payerne microwave radiometer data record (Sauvageat et al., 2023).

**Table 2. Ground-based ozone profile climate data records used in the analyses.**

|  | **Station** | **Latitude, Longitude** | **Ozone Profile Records** | **Record Length** |
|---|---|---|---|---|
| **Alpine** | Hohenpeißenberg | 47.8°N, 11.0°E | Ozonesonde | 1966–2024 |
|  |  |  | Lidar | 1987–2024 |
|  | Payerne | 46.8°N, 6.9°E | Ozonesonde | 1968–2024 |
|  |  |  | Microwave | 2000–2024 |
|  | Bern | 46.9°N, 7.4°E | Microwave | 1996–2024 |
|  | Zugspitze | 47.40°N, 11.0°E | FTIR | 2000–2024 |
|  | Arosa/Davos | 46.7°N, 9.7°E | Umkehr | 1956–2024 |
|  | Jungfraujoch | 46.5°N, 7.9°E | FTIR | 2000–2024 |
|  | OHP | 43.9°N, 5.7°E | Umkehr | 1984–2024 |
|  |  |  | Lidar | 1985–2024 |
|  |  |  | Ozonesonde | 1991–2024 |
| **Mauna Loa** |  | 19.5°N, 155.6°W | Umkehr | 1984–2024 |
| **Hilo** |  | 19.7°N, 155.1°W | Lidar | 1993–2022 |
|  |  |  | Microwave | 1995-2022 |
|  |  |  | Ozonesonde | 1982–2024 |
| **Lauder** |  | 45°S, 169.7°E | Umkehr | 1987–2024 |
|  |  |  | Lidar | 1994–2024 |
|  |  |  | Ozonesonde | 1986–2024 |
|  |  |  | FTIR | 2001–2024 |



## 2.3    CCMI model data

Here we use data from chemistry–climate models (CCMs) participating in CCMI-2022 (https://blogs.reading.ac.uk/ccmi/ccmi-2022/), which provides a new set of Chemistry-Climate Model Initiative (CCMI) community simulations. Aiming at updating projections of ozone recovery, new sets of forcings have been used. They are specified according to recommendations from WMO (2018) and the newly developed Coupled Model Intercomparison Project Phase 6 (CMIP6, https://wcrp-cmip.org/cmip-phases/cmip6/, Eyring et al., 2016) using the Shared Socioeconomic Pathways (SSPs) scenarios. A baseline scenario (refD2) has been developed that closely follows the specifications of the SSP2-4.5 pathway of CMIP6 (O'Neill et al., 2016), which is based upon current GHG scenarios with an intermediate future projection, and with ODSs from WMO (2018). The ocean conditions are either modelled (from a separate climate model simulation) or internally generated (in the case of fully ocean-coupled models). The simulation includes state-of-knowledge historic forcings, with recommendations for the 11-year solar cycle and aerosol forcings (specified as for CMIP6), with the quasi-biennial oscillation (QBO) forcing either internally model-generated or nudged from an external dataset. A summary of the experiments can be found in the July 2021 SPARC Newsletter (No. 57, https://www.aparc-climate.org/wp-content/uploads/2021/07/SPARCnewsletter_Jul2021_web.pdf).

This scenario (refD2), which provides a seamless simulation running from 1960 to 2100, was selected to compile a reference model dataset for comparisons with observations. Compared to GB22, the changes are in the models, in GHG and ozone scenarios, and included aerosol forcing.

## 3    The LOTUS regression model and methods for evaluation of trends

### 3.1    The LOTUS regression model and its updates

Trend analyses were performed using version 0.8.3 of the LOTUS regression model (https://usask-arg.github.io/lotus-regression/index.html, last access: 21 October 2025). This model applies a multiple linear regression framework based on the general least squares method to quantify the variability in ozone time series using several explanatory variables (proxies). The following proxies are used: QBO, El Niño-Southern Oscillation (ENSO), the 11-year solar cycle (F10.7 radio flux), stratospheric aerosol optical depth (sAOD). Compared to LOTUS version 0.8.0 used in GB22, version 0.8.3 incorporates updated data sources for the QBO and F10.7 proxies, and the updated sAOD record (see Table 3). Following the GB22 approach, independent linear trend (ILT) terms are used to assess long-term changes before and after the peak in ozone-depleting substances (ODS), defined as January 1997 and January 2000, respectively.

For all datasets, the LOTUS regression model was applied to deseasonalized monthly mean anomalies. Datasets originally provided as monthly mean concentrations/mixing ratio (ground-based data, GOZCARDS, SWOOSH, SBUV MOD and COH) were deseasonalized prior to the analysis using the 1998-2008 baseline. This deseasonalization step was omitted for the other datasets that provide de-seasonalized anomalies.  The deseasonalized anomalies were then fitted using the regression equation GB22:





$$y(z,t) = \beta_1(z,t) \cdot QBO_1(t) + \beta_2(z,t) \cdot QBO_2(t) + \beta_3(z,t) \cdot ENSO(t) + \beta_4(z,t) \cdot F10.7(t) + \beta_5(z,t) \cdot sAOD(t)$$
$$+ [\beta_6(z,t) + \beta_7(z,t)(t - t_1)] \cdot L_{pre}(t) + [\beta_8(z,t) + \beta_9(z,t)(t - t_2)] \cdot L_{post}(t) + \beta_{10}(z,t) \cdot Gap(t) + \varepsilon(z,t), \quad (1)$$

where $y(z,t)$ represents the monthly mean ozone anomaly at altitude $z$, $\beta_{1-10}(z,t)$ are the estimated regression coefficients and $\varepsilon(z,t)$ is the residual. The QBO is described by two orthogonal components, QBO$_1$ and QBO$_2$, derived from principal component analysis. The (independent) linear trend terms are defined as:

$$L_{pre}(t) = \begin{cases} 1, & t \le t_1 \\ 0, & t > t_1 \end{cases}, \quad L_{post}(t) = \begin{cases} 0, & t \le t_{12} \\ 1, & t > t_2 \end{cases}, \quad Gap(t) = \begin{cases} 0, & t \le t_1 \\ 1, & t_1 < t \le t_2 \\ 0, & t > t_2 \end{cases} \quad (2)$$

where $t_1$ and $t_2$ are 1 January 1997 and 1 January 2000, respectively.

To account for seasonal variability, two Fourier harmonics representing annual and semi-annual variations were applied only to the QBO and ENSO regression coefficients $\beta_k(z,t)$ expressed as:

$$\beta_k(z,t) = \beta_{k0}(z) + \sum_{i=1}^{2} \left[ \beta_{k1i}(z) \sin\left(\frac{2\pi i t}{12}\right) + \beta_{k2i}(z) \cos\left(\frac{2\pi i t}{12}\right) \right], \quad k = 1, 2, 3 \quad (3)$$

Autocorrelations were removed using the Cochrane-Orcutt transformation (Cochrane and Orcutt, 1949).

The LOTUS model was applied to the merged satellite ozone records from 1985-2024 across all latitude bands and altitude or pressure levels, depending on the native coordinates of the datasets. The same approach was used for each vertical level of the ground-based observations at selected NDACC stations and for the gridded satellite datasets (MEGRIDOP, SWOOSH, SCIAMACHY-OMPS and SBUV-MOD) over the regions surrounding those sites.

For the CCMI models, the trend analyses were performed over the period 1979–2024 using the LOTUS regression model. We calculated the appropriate QBO and El Niño–Southern Oscillation (ENSO) proxies from the model data (zonal winds and sea surface temperatures, SSTs) and used the external forcings (e.g. 11-year solar cycle) as provided to the modelling groups.

**Table 3. Proxy time series used in the LOTUS regression model.**

| Variable | Proxy | Source - LOTUS 0.8.3 | Source - LOTUS 0.8.0 |
|---|---|---|---|
| **QBO(t), QBO2(t)** | Two orthogonal components of the QBO calculated using principal component analysis | https://acd-ext.gsfc.nasa.gov/Data_services/met/qbo/QBO_Singapore_Uvals_GSFC.txt (last access: 21 August 2025) | http://www.geo.fu-berlin.de/met/ag/strat/produkte/qbo/qbo.dat (last access: 22 June 2022) |



| Solar(t) | Solar 10.7 cm flux | https://spdf.gsfc.nasa.gov/pub/data/omni/low_res _omni/omni2_all_years.dat  (last access: 21 August 2025) | https://spaceweather.gc.ca/forecast-prevision/solar-solaire/solarflux/ sx-5-mavg-en.php (last access: 22 June 2022) |
|---|---|---|---|
| ENSO(t) | Multivariate El Niño–Southern Oscillation (ENSO) index without lag (MEIv2) | https://psl.noaa.gov/enso/mei/  (last access: 21 August 2025) | https://psl.noaa.gov/enso/mei/ (last access: 22 June 2022) |
| sAOD(t) | GloSSAC stratospheric aerosol optical depth | https://asdc.larc.nasa.gov/project/GloSSAC/GloS SAC_2.21 (last access: 21 August 2025) | https://asdc.larc.nasa.gov/project/GloSSAC/Glo SSAC_2.0 (last access: 22 June 2022) |

## 3.2    Evaluation of multi-dataset mean trends and its uncertainty

### 3.2.1    The mean trends for merged satellite and ground-based data

The evaluation of the mean (overall) trend from the satellite datasets follows the approach described in GB22 and P19. First, the trends from different datasets are interpolated to the same vertical grid. The combined trend $\bar{x}$ is evaluated as the arithmetic mean of the trends from individual merged datasets.

The trends from various datasets and their uncertainties cannot be considered as independent random variables: correlations of uncertainties are caused by using data from the same satellite instruments (for example, SAGE II data are used in all merged

datasets except for SBUV MOD and COH; MLS data are used in both GOZCARDS and SWOOSH etc, see Table 1), and also from the natural ozone variability (common in all datasets) which is not captured by the regression model. The uncertainty of combined trends is estimated as (GB22, P19):

$$\sigma^2_{comb} = \max\left( \frac{1}{N^2} \sum_{i,j} C_{ij} \sigma_i \sigma_j, \frac{1}{n_{eff}} \sum \frac{(x-\bar{x})^2}{N-1} \right), \qquad (4)$$

where $N$ is the number of observation records, $C_{ij}$ are the correlation coefficients for the trend estimates $x_i$ from data sets $i$

and $j$ (see also below), $\sigma_i$ are the trend uncertainties estimated from the fit residuals for the individual data sets, and $n_{eff}$ is the effective number of independent trend estimates.

The first term in Eq.(4) is the variance of the mean of correlated values, obtained through traditional propagation of errors. It serves as an approximation of the theoretical lower bound of trend uncertainty due to the actual realization of the ozone time series. The second term is the unbiased estimator of the standard error of the mean. It can capture biases in trend uncertainties





between the different merged data sets that would not be captured by the first term (e.g., resulting from drifts between data

sets or differing unit representations). The effective number of independent values $n_{eff}$ in Eq. (4) is approximated by

$$n_{eff} = \frac{N^2}{\sum_{ij} C_{ij}}$$
(5)

In Eq. (4), we do not use the sum of both terms because the variance of the trend estimates (term 2) can be partly due to the uncertainties represented by term 1. In the upper stratosphere, the second term usually dominates, while these terms have

210 comparable values in the lower stratosphere (P19 and Figure S6).

Correlation coefficients of trend uncertainties are approximated by the correlation coefficients of fit residuals from the regression model. P19 discusses in detail the challenges of estimating the correlation of trend uncertainties, as these correspond to the largest temporal scales in the ozone time series. This approach has limitations, and the estimated correlations correspond to the upper bound of the true correlations of trend uncertainties. The values of the correlation matrix are explicitly written in

the Supplement.

### 3.2.2    The mean trends for the models

The calculation of the CCMI-2022 multi-model mean trends follows the approach described in GB22 and P19. First, ozone data from each model were interpolated to a common zonal bin (5° latitude). Individual trends were then calculated for all models at their given pressure levels and the 5° latitude bins. In the case of multiple simulations performed by a model, the

220 ensemble mean trend was calculated.

Results from the final set of models (9 in total, including ensemble means) were averaged over the appropriate latitude bands (60°S - 35°S, 20°S - 20°N, 35°N - 60°N), as the mean of individual model trends, with the uncertainties calculated with error propagation.

## 4    Trend results and discussions

## 4.1    Zonal mean trends

Trends in vertical ozone profiles as a function of latitude are estimated using the eight merged satellite datasets with the LOTUS regression model as described in Sect. 3.1.  For all datasets, the trends are evaluated using the period from 1985 to 2024. Figure 1 shows post-2000 trends as a function of latitude and pressure/altitude. The horizontal resolution and the vertical coordinates correspond to the original dataset grids. The black lines in Figure 1 show the mean climatological thermal



tropopause height (or pressure) from the ERA-5 reanalysis estimated using the post-2000 period. The trends in the upper troposphere are shaded and not assessed in this study.

For all merged datasets, positive trends 1-4 % decade$^{-1}$ are observed in the upper stratosphere, they are statistically significant at 95% confidence level (this confidence level is used in all evaluations of statistical significance in our paper). The upper stratospheric trends in the Northern Hemisphere (NH), above ~35 km, are larger by 1-2% for SAGE-OSIRIS-OMPS
and SAGEII-OSIRIS-SAGEIII than for other merged datasets. In the lower stratosphere (~7-10 km above the tropopause), nearly all datasets report negative trends in majority of latitude zones, from -1 to -6% decade$^{-1}$(depending on latitude), but the uncertainties of trend estimates are large, so that they are not significant (indicated by gray stippling in Fig. 1).

The morphology and magnitude of ozone trends have not changed significantly compared to the previous analyses reporting 2000-2020 trends (compare with Figure 3-10 in WMO-2022, adapted from GB22, Fig.1). For quantitative
comparisons, we need to keep in mind that trend changes can be due to (i) extension of time period (from 2020 to 2024), (ii) changes in the datasets (versioning) and (iii) changes in the trend regression model. To address these issues, we carried out sensitivity studies comparing ozone trends across old and new datasets, trend model setups, and time periods.

We found that updated proxies in the regression model have a small impact on ozone trends: the differences are generally less than 0.5% per decade with larger local deviations (up to 2% per decade) in the UTLS (Figure S1).
The changes in 2000-2020 trends evaluated using old (as in GB22) and new versions of the merged satellite datasets are illustrated in the Supplementary Figure S2. In the middle and upper stratosphere, the changes in ozone trends due to updates in the datasets are typically less than 0.5 - 1%. In the UTLS, especially in the tropics, the changes are larger: the trends became less negative for SAGE-CCI-OMPS+, SAGE-OSIRIS-OMPS, SAGE-SCIAMACHY-OMPS, likely due to changes in OMPS-LP version. However, the uncertainties of ozone trends in the UTLS exceed the trend estimates, so all these changes are smaller
than the trend uncertainties.

The influence of the 4-year time extension on ozone trends is illustrated in Supplement Figure S3, which compares 2000-2024 and 2000-2020 trends for individual merged satellite datasets. All datasets show a slight increase in positive trends by ~1% per decade in the middle stratosphere (10-30 hPa), mainly in the tropics and Northern Hemisphere. Some of the datasets show a slight decrease of trends in the SH upper stratosphere, by 0.5-1%. All these changes are within uncertainty
intervals of the trends.

The year 2024 was characterized by high ozone values in the middle and lower stratosphere at NH mid- and high latitudes (Newman et al., 2024). When excluding this year from the time series and evaluating the trends until 2023, the ozone trends are less positive in the NH middle and low stratosphere (Figures S4 and S5 in the Supplement). Figure S4 also illustrates the sensitivity of trend estimates to large changes in time series at the endpoint. These changes are also within uncertainty
intervals for trends



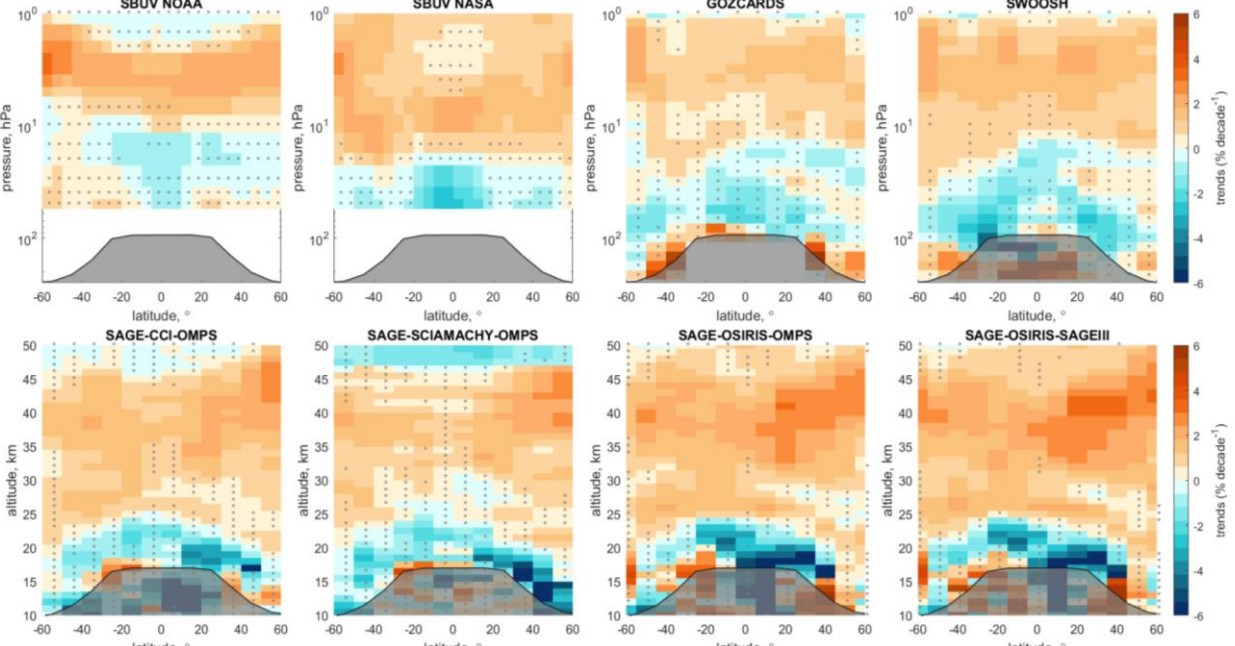

**Figure 1. Ozone trends (% decade⁻¹) for the period 2000–2024 estimated from eight merged satellite data records using an independent linear trend model. Trends are shown for the SBUV COH (NOAA), SBUV MOD (NASA), SWOOSH, GOZCARDS, SAGE-CCI-OMPS, SAGE-SCIAMACHY-OMPS, SAGE-OSIRIS-OMPS, and SAGEII-OSIRIS-SAGEIII datasets. Gray stippling**
**denotes results that are not significant at the 2σ level. Data are presented on the vertical coordinates (left-hand axis) and latitudinal grid associated with each dataset. The black lines show the mean climatological thermal tropopause height (or pressure) from the ERA-5 reanalysis . The trends in the upper troposphere are shaded and not assessed in this study.**

Figure 2 presents the ozone profile trends in DU km⁻¹ decade⁻¹, for the same merged satellite datasets as in Figure 1. The trends
in absolute values are obtained from percentage trends using the climatological ozone distribution from the SAGE-CCI-OMPS+ dataset. Positive and statistically significant trends are observed in the upper stratosphere (Figs. 1-2). However, since the peak ozone abundance is in the layer ~5-15 km above the tropopause, the contribution of these positive trends to changes in the total ozone column is relatively small.  Ozone trends are mostly negative in the lower stratosphere (the ~8-10 km layer above the tropopause), on the order of −0.2 to −0.4 DU km⁻¹ decade⁻¹, and mostly positive above this layer in the middle
stratosphere, at about 0.2 to 0.4 DU km⁻¹ decade⁻¹. The magnitude and significance of these lower- and middle-stratospheric trends vary across the datasets, with several merged datasets (e.g., SWOOSH, SAGE-CCI-OMPS+, SAGE-OSIRIS-OMPS) reporting slightly stronger positive trends in the Southern Hemisphere middle stratosphere.




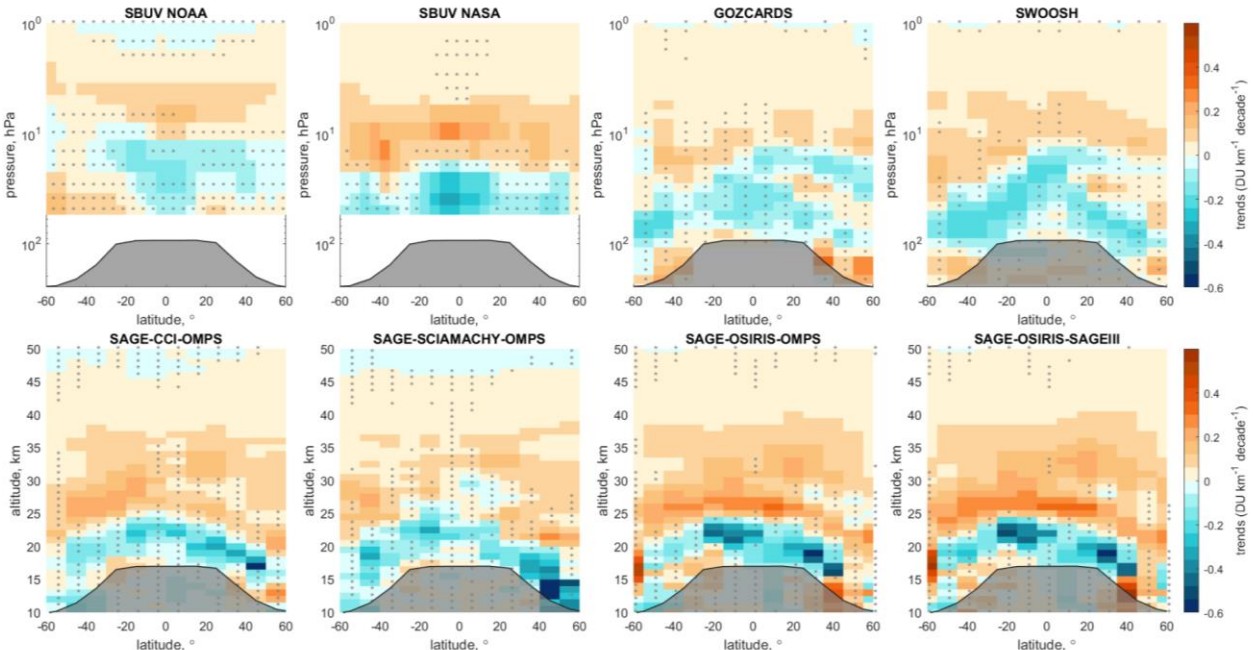

**Figure 2. Same as Figure 1, but trends are expressed in DU km⁻¹decade⁻¹.**

Figure 3 shows the combined (mean) trend derived from eight merged datasets as described in Section 3.2.1, in units of % decade$^{-1}$ (panel a) and DU km$^{-1}$ decade$^{-1}$ (panel c), with indication of its statistical significance. The combined trends shown in Fig. 3 capture the patterns in ozone profile trends that are common across the eight individual datasets analyzed in this study. Significant positive trends are observed in the upper stratosphere (35–45 km), reaching about 2–2.5% decade$^{-1}$ (Fig. 3a), with stronger trends at mid-latitudes. In the middle stratosphere (25–35 km), the trends are also positive but not statistically significant. In contrast, trends in the lowermost stratosphere (~10–18 km in the mid-latitudes and ~17–25 km in the tropics) are mostly negative, though not always significant.

When the combined trends are expressed in DU km$^{-1}$ decade$^{-1}$ (Fig. 3c), the strongest positive trends of approximately 0.1 DU km$^{-1}$ decade$^{-1}$ occur in the middle stratosphere (25–35 km). Positive trends are also present in the upper stratosphere (above 35 km), but their magnitudes in DU km$^{-1}$ units are relatively small. Negative trends in the lower stratosphere are more pronounced, reaching about –0.2 DU km$^{-1}$ decade$^{-1}$, but they are generally not statistically significant at 95 % confidence level.





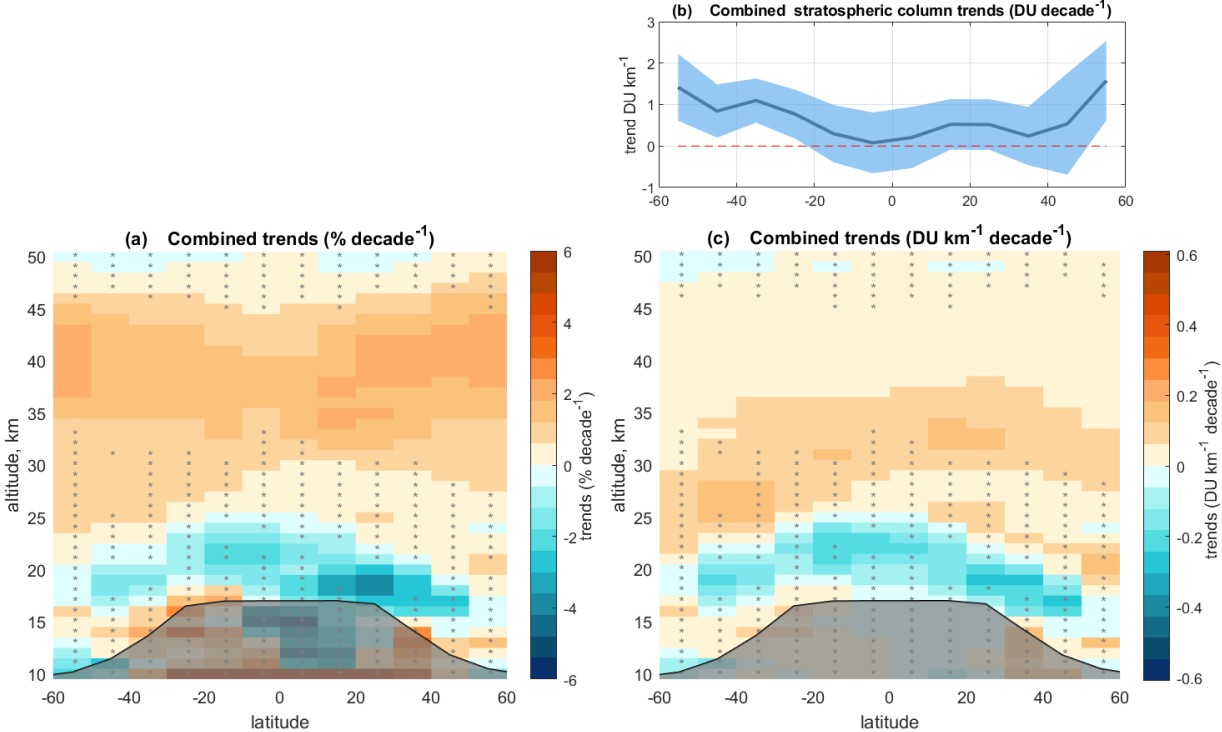

**Figure 3. The combined (mean) ozone trends from eight merged satellite datasets as a function of latitude and altitude, expressed in % decade[-1] (panel (a)) and DU km[-1] decade[-1] (panel(c)). The black lines show the climatological mean tropopause height. Gray stippling denotes results that are not significant at the 2σ level. Panel (b): stratospheric column averaged trend (DU decade[-1]), blue line, with 2σ uncertainty (blue shading). Red dashed line in panel (b) highlights zero level.**

The net trend in the stratospheric ozone column (Fig. 3b) is generally positive but not statistically significant, except for the SH mid-latitudes (20°S–60°S) and the NH high latitudes (50°N–60°N), where trends are significant and reach about 1 DU decade[-1]. In the tropics (20°S–20°N) and NH subtropics (20°N–40°N), the net stratospheric column trends are smaller, around 0.1–0.5 DU decade[-1], and not statistically significant. The latitudinal dependence of the stratospheric ozone column trend is similar to that of total ozone column trend (WMO-2022, Weber et al., 2022).

For evaluation of trends in broad latitudinal bands (35-60°S, 20°S-20°N, 35°-60°N), we used the same seven merged datasets as for GB22. The deseasonalized anomalies were first averaged in broad zones, and then the LOTUS regression model was applied. Ozone trends from individual merged datasets in broad zones are shown in Figure 4, together with the combined (mean) trend evaluated as described in Section 3.2.1. All datasets report positive and statistically significant trends ~2% decade[-1] in the upper stratosphere, with stronger trends at mid-latitudes. In the tropical lower stratosphere, all datasets report negative trends, but with large uncertainties. At mid-latitudes, the lower stratospheric trends are mostly negative with large uncertainty intervals that encompass zero trends.





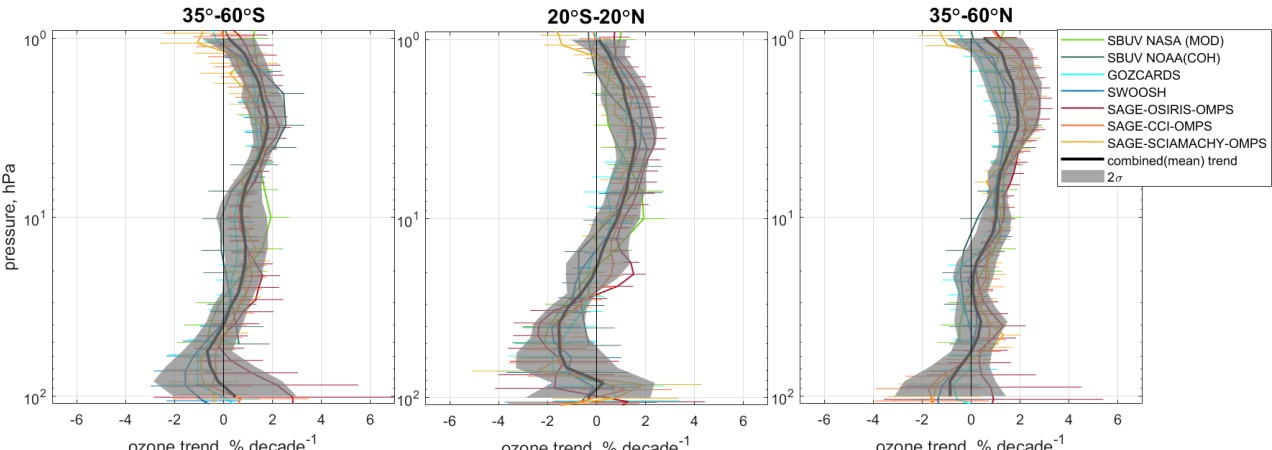

**Figure 4. Ozone profile trends with 2σ uncertainties for the period 2000–2024 for latitude bands 35–60°S (left panel), 20°S–20°N (center panel), and 35–60°N (right panel). Colored lines are the trend estimates from seven individual merged datasets on their original vertical grids (SBUV NASA (MOD), SBUV NOAA (COH), GOZCARDS, SWOOSH, SAGE-OSIRIS-OMPS, SAGECCI-OMPS, and SAGE-SCIAMACHY-OMPS). Black lines represent the mean (combined) trends and gray shading indicates the 2σ uncertainty intervals for the combined trends.**

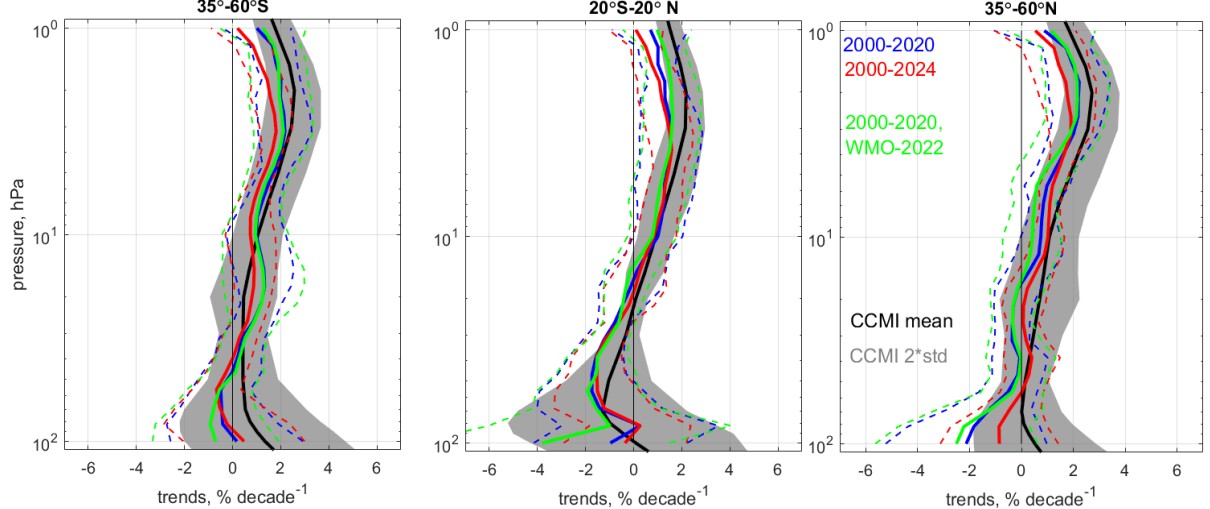

**Figure 5 Comparison of simulated and observed post-2000 ozone trend profiles for the latitude bands 35–60°S (left panel), 20°S–20°N (center panel), and 35–60°N (right panel). Observed trends for the period 2000-2020 from GB22 and WMO-2022 is shown in green and the results with the updated regression model and updated datasets for the period from 2000 to 2020 and to 2024 are shown in blue and red, respectively. The mean model trends (CCMI REF-D2, 9 models) for 2000-2024 are shown in black with the 2σ uncertainties shown as shaded grey areas.**

Figure 5 compares the mean ozone trends in broad latitude zones for 2000-2020 and 2000-2024 using seven merged satellite datasets (as in Figure 4). Overall, mean trends in ozone profiles are highly consistent with the trends calculated for 2000-2020



(compare red and blue lines in Figure 5). At mid-latitudes, a small reduction of the positive trends in the upper stratosphere by ~0.5% decade[-1] is observed (above 10 hPa in SH and above 3 hPa in NH), though these reductions are within the estimated uncertainties. In the NH mid-latitudes, the 2000-2024 trends become more positive in the middle stratosphere (7-20 hPa) and less negative or neutral in the lower stratosphere (100-30 hPa). As discussed earlier, this is primarily due to the high ozone levels observed in NH in 2024 (Newman et al., 2024). Tropical trends remain nearly unchanged. The uncertainties in the observed mean trends are slightly smaller for the 2000-2024 period, as expected due to the extended time period (see Supplement Figure S6 for more details).

The 2000–2020 trends shown here (blue lines) have changed slightly compared to those presented in GB22 (green lines), which were also calculated over the same period. Less negative trends, on the order of 0.5–1% decade[-1], are observed in the UTLS, for all latitude zones. These changes are primarily due to updates in the merged satellite ozone datasets (see Table 1) and they are consistent with the trend changes in 10° latitude bands discussed above (Figure S2). There are no notable changes in the middle and upper stratosphere as a result of these dataset updates.

Figure 5 also shows ozone trends for 2000-2024 predicted by REF-D2 CCMI simulations (black lines for multi-model mean and grey shading for 2σ uncertainties). The mean CCMI ozone trends are very close to the mean satellite trends in the middle stratosphere, for all broad latitude zones. In the upper stratosphere, models predict a slightly stronger ozone recovery of +2-2.5% decade[-1] compared to the value from satellite observations of ~+1.5 % decade[-1]. In the tropical lower stratosphere, both models and satellite observations report negative trends of ~ -1 %decade[-1], but the shape of the profile trend is slightly different (but within estimated uncertainty intervals). At mid-latitudes, simulated and observational ozone trends in the lower stratosphere are now closer to each other than reported in previous assessment (compare with Figure 5 in GB22).

### 4.2    Trends over selected NDACC stations

Figure 6 shows the comparison between ground-based ozone trends from NDACC stations and those from combined satellite data records at three locations: (a) the SH mid-latitude site at Lauder, New Zealand; (b) the tropical sites at Hilo and Mauna Loa, Hawaii, USA; and (c) the NH mid-latitude Alpine region. In order to compare with ground-based records, we selected the MEGRIDOP, SWOOSH, SCIAMACHY-OMPS, and SBUV MOD satellite data from the nearest grid cell to the station location. Then the satellite trends were averaged to obtain a combined trend as described in Sect. 3.2.1. Alpine ground-based trends were calculated as the mean of the following records: 1) ozonesondes from Hohenpeißenberg, Payerne, and Haute Provence; 2) lidar from Hohenpeißenberg and Haute Provence; 3) Umkehr from Arosa/Davos and Haute Provence; 4) FTIR from Zugspitze and Jungfraujoch; 5) MWR from Payerne and Bern.

It should be noted that the FTIR ozone profiles have a relatively low-vertical resolution (only about 3 Degrees of Freedom for Signal in the stratosphere (Vigouroux et al., 2015)), therefore the FTIR trends plotted in Fig. 6 represent ozone changes in wide partial columns 12-20 km, 20-29 km, and 29-48 km.



Overall, the agreement between the ground-based and combined satellite trends is good, the trends mostly remain within uncertainty intervals. Note that the spatio-temporal sampling patterns are different for ground-based and satellite measurements, thus contributing to trend differences. The largest discrepancies in trends from different GB instruments at a single location occur at Lauder.

The Lauder station (Fig. 6a) shows negative trends of up to -2.5-6% decade$^{-1}$ in the lower stratosphere (below 20-25 km) and
positive trends of up to 1.3% decade$^{-1}$ in the upper stratosphere (above about 35 km). The trends from Dobson Umkehr and lidar data agree well with the combined satellite trends, while ozonesondes show stronger decline in the lower stratosphere. Ozonesonde trends are outside the satellite mean uncertainty range. A similar discrepancy was reported in GB22 (Fig.4 in GB22) with even stronger negative trends from ozonesondes at Lauder.

At the Mauna Loa/Hilo site (Fig. 6b), positive trends of up to 1.5% decade$^{-1}$ are observed in the upper stratosphere, while
negative trends appear in the lower stratosphere (up to -5-6% decade$^{-1}$). The instruments are largely consistent, though the lidar measurements show smaller positive or even slightly negative trends in the middle stratosphere (25-35 km). Note that the Mauna Loa lidar and MWR data records end at the end of 2022 due to the Mauna Loa volcanic eruption.

The Alpine stations (Fig. 6c) show strong positive trends in the upper stratosphere, reaching +2.5-4% decade$^{-1}$, and negative trends of up to -4% decade$^{-1}$ in the lower stratosphere. Above 45 km, the trends remain positive but decrease. Overall, the
Alpine sites exhibit the strongest positive trends in the upper stratosphere and the weakest negative trends in the lower stratosphere among all stations. The agreement across different instrument techniques is the best among the three selected study cases.

In general, for all stations, the trends from ground-based measurements are consistent with the satellite mean trends (black line) and capture the same overall vertical pattern of ozone trends. They show ozone recovery in the upper stratosphere, weaker
or transitioning trends from negative to positive in the middle stratosphere, and negative trends in the lower stratosphere. Compared to Fig.4 in GB22, the agreement between ground-based and satellite measurements has improved (e.g., for the FTIR at Lauder), but there are still differences (e.g., ozonesonde trends at Lauder and Umkehr trends in various locations in the lowermost stratosphere). The uncertainties of the trends from ground-based instruments are also reduced uncertainties compared to GB22, particularly in the lower stratosphere. Trends in the upper stratosphere differ only slightly (<1% decade$^{-1}$
), while the lower stratospheric ground-based trends are less negative compared to GB22. These changes remain within the ground-based trend uncertainties.

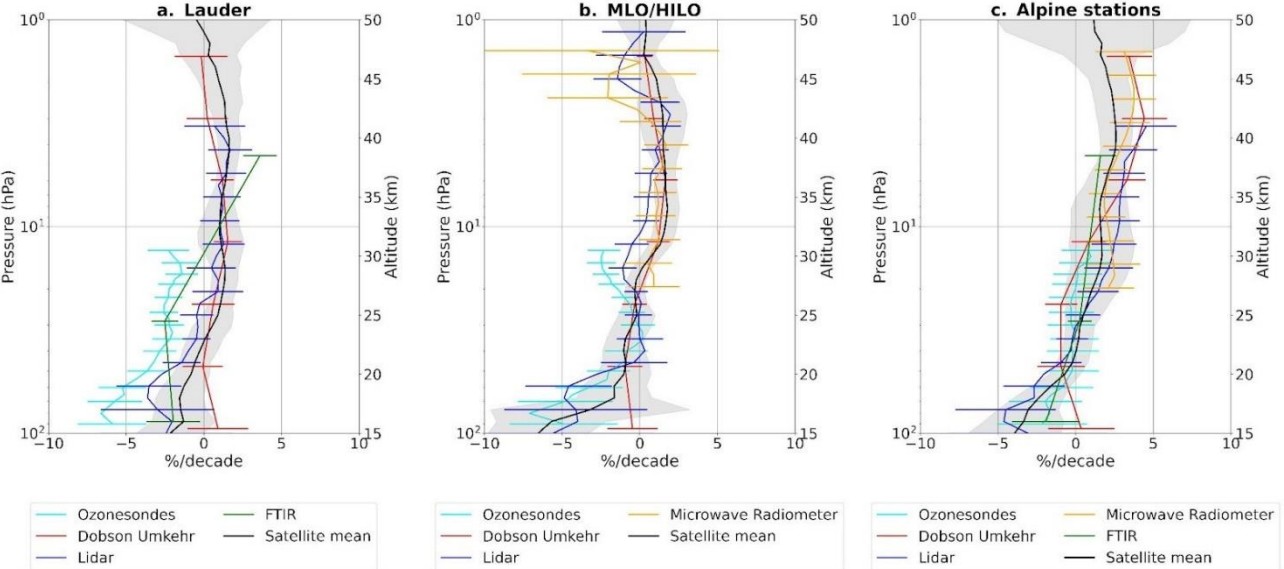

**Figure 6. Ozone trend profiles (%/decade) for 2000–2024 at selected NDACC locations: a. Lauder (New Zealand); b. Mauna Loa and Hilo (Hawaii, USA); c. Alpine stations that include (1) ozonesondes from Hohenpeißenberg, Payerne, and Haute Provence; (2) lidars from Hohenpeißenberg and Haute Provence; (3) Umkehr from Arosa and Haute Provence; (4) FTIR from Zugspitze and Jungfraujoch; (5) MWR in Payerne and Bern. Trend profiles are shown for ozonesondes (light blue), Dobson Umkehrs (red), lidars (dark blue), microwave radiometers (orange), and FTIR (green). Also shown is the mean satellite trend profile, estimated by averaging MEGRIDOP, SWOOSH, SCIAMACHY-OMPS, and SBUV-MOD satellite data from the nearest grid boxes. Error bars represent the 95% confidence intervals**

## 4.3 Global regional trends

For evaluation of regional trends, we used three merged altitude-gridded datasets with resolved longitudinal structure: MEGRIDOP, SCIAMACHY-OMPS and SWOOSH. Figure 7 shows the horizontal structure of ozone trends in the period 2003-2024 at 5 altitude levels: 20, 25, 30, 35, and 40 km. In addition to aforementioned datasets, we also show trends from the MLS dataset (evaluated using the period 2004-2024). The trends are evaluated using the same setup of the multiple linear regression model, but with a single linear trend term due to shorter time period.

A strong longitudinal dependence of trends in NH extratropical ozone (at latitudes poleward 50°N) trends and below 40 km is observed. The trends are characterized by a dipole-like structure with positive trends over Scandinavia and negative trends over Siberia, which had been reported by Arosio et al. (2019) and Sofieva et al. (2021) based on 2003-2018 time period. Arosio et al (2024) has shown that model simulations also reproduce this trend pattern in ozone trends. The study concludes that this zonal asymmetry is primarily driven by dynamical processes, such as a weakening and shift in the activity of planetary wavenumber-1 over the past two decades (Arosio et al., 2024).



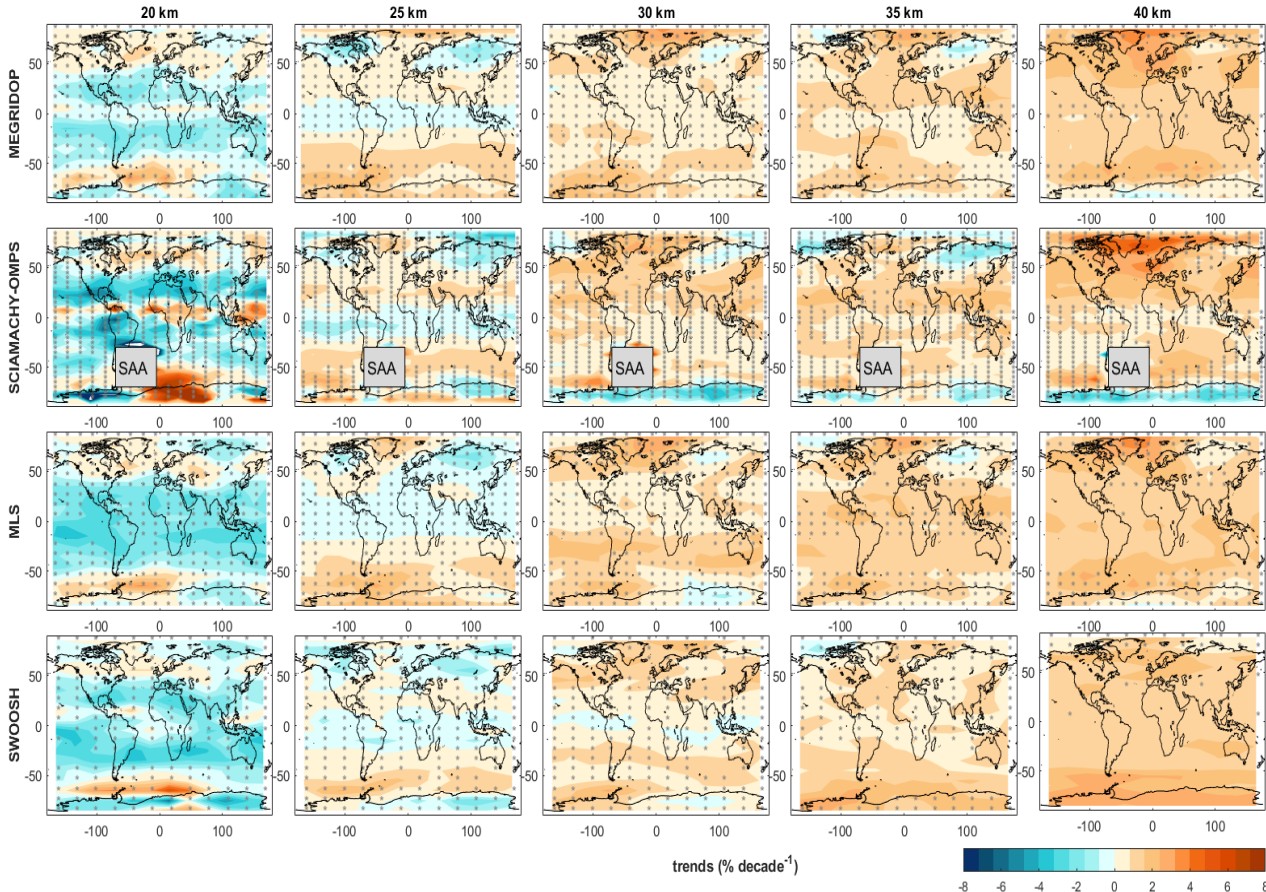

**Figure 7. Latitude- and longitude-dependent ozone trends (% decade⁻¹) derived for the period 2003–2024 for five different altitude**
**levels, based on the extended MEGRIDOP, SCIAMACHY-OMPS, SWOOSH, and MLS datasets. Gray dots indicate regions where the trends are not statistically significant at the 2σ level. For the SCIAMACHY-OMPS dataset, ozone trends in the Southern Atlantic Anomaly (SAA) region are not shown because SCIAMACHY data are flagged in this region.**

The magnitude of both positive and negative trends in the dipole-like patterns over the NH has slightly decreased compared to
415 the earlier studies by Arosio et al. (2019) and Sofieva et al. (2021). At the same time, lower stratospheric (around 20 km) trends in the SH (20°S–55°S) are more zonally uniform now with predominantly negative but insignificant trends. The patchy positive values in the SCIAMACHY+OMPS data set in the inner-tropical lower stratosphere are caused by Hunga-Tonga aerosol-related artifacts in the UV/VIS satellite ozone retrieval.

A small negative trend has emerged over Antarctica above 40-45 km (seen especially in SCIAMACHY-OMPS); however, this
negative trend is not statistically significant. Since natural variability is high in polar regions, it is quite expected that a simple



multiple regression gives trend estimates that are not statistically significant and sensitive to selected (or available) time period. Other methods for trend analysis in polar regions, such as considering seasonal trend (Galytska et al., 2019; Solomon et al., 2016; Szeląg et al., 2020), using dynamical proxies (e.g., Weber et al., 2022) or vortex-related coordinates can be explored in future works.

## 5    Summary

In this paper, we presented an updated analyses of trends in stratospheric ozone profiles in the latitude range 60°S-60°N using long-term ground-based and merged satellite climate data records, including a comparison with chemistry-transport models.

The ground-based datasets include the climate data records constructed from ozonesonde, lidar, FTIR, Dobson Umkehr, and microwave radiometer data. Satellite climate data records include eight merged datasets with zonally averaged profiles: SBUV MOD, SBUV COH, GOZCARDS, SWOOSH, SAGE-CCI-OMPS+, SAGE-SCIAMACHY-OMPS, SAGE-OSIRIS-OMPS and SAGEII-OSIRIS-SAGEIII. In addition, regional trends are estimated using the merged satellite datasets with resolved longitudinal structure: MEGRIDOP, SCIAMACHY-OMPS, and SWOOSH. Most datasets have been updated since GB22 and WMO-2022 by incorporating new versions of existing satellite datasets, adding new satellites, and other developments.

The trends in ozone profiles have been evaluated using the LOTUS multiple linear regression model with QBO, ENSO, solar flux and aerosol proxies and independent-linear term for trends. The whole time series starting from 1985 are used for trend evaluation, but only post-2000 trends are discussed in this paper. The method of Petropavlovskikh et al. (2019) is applied to estimate the overall combined trends and uncertainties from the satellite data. The paper presents a detailed characterization of global (zonally averaged and in broad latitude zones) and regional (with resolved longitudinal structure) ozone trends, in both relative and absolute units, from individual climate data records as well as the overall combined trends.

Overall, the mean trends in ozone profiles over 2000-2024 are highly consistent with the trends calculated over 2000-2020 and reported in WMO-2022 and references therein. Our analyses of satellite data confirm the statistically significant positive ozone trends in the upper stratosphere ~1-3 % decade$^{-1}$, with larger trends at mid-latitudes compared to the tropics. Compared to WMO-2022, the upper stratospheric trends are nearly unchanged in the tropics but are slightly reduced at mid-latitudes. The 2000-2024 trends in the NH middle stratosphere are more positive compared to the 2000-2020 trends, partly due to the high NH ozone levels in 2024. In the lower stratosphere, the trends are mostly negative, -1-2 % decade$^{-1}$. Compared to the 2000-2020 period, the lower stratospheric trends are nearly unchanged in the tropics and SH mid-latitudes, but they became less negative at NH mid-latitudes.

Ozone trends predicted by REF-D2 CCMI simulations in the period 2000-2024 are in very close agreement with the mean satellite trends in the middle stratosphere, for all broad latitude zones. In the upper stratosphere, models predict a slightly stronger ozone recovery of +2-2.5 % decade$^{-1}$ than observations. In the lower stratosphere, both models and satellite



observations report negative tropical trends of ~ -1 % decade$^{-1}$, while modelled ozone trends are slightly positive at mid-latitudes. However, simulated and observational ozone trends in the mid-latitude lower stratosphere are now closer to each other than reported in GB22 and WMO-2022. All estimated trends in the lower stratosphere have large uncertainties, and they are not statistically significant.

Ozone profile trends over several stations estimated from ground-based records generally agree with the trends estimated using merged satellite datasets with resolved longitudinal structure. We found a good agreement between trends from ground-based and satellite measurements in the tropics and in the Alpine stations, while trend values for different types of ground-based instruments are more scattered at the Lauder station (but nevertheless improved compared to WMO-2022).

The analysis of regional ozone profile trends in 2003-2024 using merged satellite datasets confirmed the previous observations of a longitudinal structure in ozone trends in the NH mid-latitude stratosphere, with positive trends over Scandinavia and negative trends over Siberia. However, the magnitude of this dipole-like structure is reduced due to extension of the data records. The estimated regional trends are consistent with the zonally averaged trends.

**Data availability[1]**

All merged satellite datasets are available at LOTUS ftp website (contact the first author for access). In addition, they are available from open access sources listed below.

SAGE-CCI-OMPS+, SAGE-SCIAMACHY-OMPS, MEGRIDOP and SCIAMACHY-OMPS datasets are available at https://climate.esa.int/en/projects/ozone/data .

SBUV-COH v8.6 is available at https://www.ftp.cpc.ncep.noaa.gov/SBUV_CDR .

SBUV-MOD v8.7 is available from http://acdb-ext.gsfc.nasa.gov/Data_services/merged/ .

GOZCARDS is available at https://search.earthdata.nasa.gov/ and the LOTUS ftp.

SWOOSH is available at https://csl.noaa.gov/groups/csl8/swoosh/ .

SAGE-OSIRIS-OMPS and SAGEII-OSIRIS-SAGEIII are available at https://research-groups.usask.ca/osiris/data-products.php#OSIRISLevel3andMergedDataProducts .

The Jungfraujoch FTIR ozone data (DOI: 10.60897/ndacc.irwg2023.o3.jungfraujoch.R001) are available at NDACC: https://www-air.larc.nasa.gov/pub/NDACC/PUBLIC/stations/jungfraujoch/hdf/ftir/ and EVDC: https://evdc.esa.int/doi/10.60897/ndacc.irwg2023.o3.jungfraujoch.R001

The Mauna Loa lidar data are available at NDACC: https://www-air.larc.nasa.gov/missions/ndacc/data.html?station=mauna.loa.hi/hdf/lidar/

The Bern, Mauna Loa and Payerne MWR data are available at NDACC https://ndacc.larc.nasa.gov/instruments/microwave-radiometer.

---

[1] At the stage of paper acceptance, all trend results will be provided in open-access repository



**Author contributions**

VS and MS performed trend analyses of satellite and ground-based data, prepared illustrations and wrote the main part of the paper. KT provided the trends from CCMI simulations. DZ, KD, and DD provided and maintained the LOTUS regression model. Other co-authors contributed with satellite or ground-based data. The results of the study were discussed by all co-authors, and all co-authors contributed to writing the paper.

**Acknowledgements**

Much of the ground-based data used in this publication are part of the Network for the Detection of Atmospheric Composition Change (NDACC) and are available through the NDACC website www.ndacc.org. The author thank the HEGIFTOM team for homogenizing ozonesonde data.

Brian Auffarth, Alexei Rozanov, Mark Weber, Carlo Arosio, Viktoria Sofieva, and Kleareti Tourpali acknowledge support from ESA Contract No. 4000137112/22/I-AG "Ozone Recovery from Merged Observational Data and Model Analysis 495 (OREGANO)".

J.D. Wild is supported by NOAA grant NA24NESX432C0001 (Cooperative Institute for Satellite Earth System Studies - CISESS) at the University of Maryland/ESSIC.

Work performed at the Jet Propulsion Laboratory, California Institute of Technology, was done under contract with the National Aeronautics and Space Administration. The authors acknowledge the work of John Anderson for HALOE data 500 updates and Ray Wang for SAGE-I and SAGE-II data updates and the production of GOZCARDS version 2.2 merged data.

Glen McConville, Peter Effertz and Irina Petropavlovskikh were supported by NOAA Cooperative Agreement with CIRES, NA17OAR4320101.

Stacey Frith is supported by NASA programmatic fund "Long-term ozone trends" (project no. WBS 479717).

The Jungfraujoch FTIR monitoring program was primarily supported by the F.R.S. - FNRS (Brussels, Belgium), the GAW-505 CH program of MeteoSwiss (Zürich, Switzerland) and the University of Liège, Belgium. E. Mahieu is a research director with F.R.S. - FNRS.

OHP ozone lidar and sonde measurements and analysis are supported by funding from CNRS Earth & Space.



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
