# Peer review of "Updated global and regional trends of stratospheric ozone profiles"

_EGUsphere, 2025_

## Referee Comment (RC2)

***Review of egusphere-2025-5963, "Updated global and regional trends of stratospheric ozone profiles", by Sofieva et al.***

This paper is very well written and presents analyses of changes in the abundance of stratospheric ozone as a function of pressure, for various geographic regions. The results are important, and the paper is an essential element of the upcoming Scientific Assessment of Ozone Depletion report. In my opinion, the paper should be accepted after consideration of the following comments.

**Major comments:**

While I like the fact that the paper is relatively brief, and I especially like the fact that the paper is so well-written, the choice for brevity means the reader has to go elsewhere for important details. A lot of these comments, which together I deem to be "major", represent areas where the paper could be improved by providing information I think is important enough to warrant an explanation.

Line 111: Table 1 gives 10 so-called "merged satellite datasets". However, only 8 of these 10 are used for the bulk of the paper. I did no not recall reading any words about why 2 datasets appear in Table 1, and are then largely not used. Also, in Table 1, the SWOOSH dataset is one of three used for assessing ozone trends as a function of latitude and longitude. For the other two of these datasets, the longitudinal resolution of the data are given in the second column of this table. Therefore, the longitudinal resolution of SWOOSH should be added. Finally, the last column of Table 1 is very important, but lacking in detail. For instance, for SWOOSH, the table states "Updated version of MLS". For SAGEII-OSIRIS-OMPS, we see "New dataset". Nearly every entry in this last column is lacking vitally important detail. Here, for all datasets, need either the dataset version number, or if this is not possible to provide, some data marker for either when the data were last accessed or even better, when the retrievals were last updated. P.S. I know some of this info is in the text. Still, should repeat in the table in my opinion.

Lines 103 to 104: Sentences "In GOZCARDS ... after May 31, 2024)." Need some work. No need to end a sentence with the word "this". I'd end the first sentence with "used prior to WMO 2022" and strike "In fact" from the start of the second sentence.

Line 109 to 110: unclear which "ozone profiles" where examined by Arosio et al. (2024); please add more detail here.

Line 116": can strike "the" just before "access"

Line 160: So, the regression considers two time periods: before January 1997 and after January 2000. I know of course this is decision of the LOTUS team. Here, or elsewhere in the paper, I think the reader would like to know: a) what happens to ozone measurements obtained from February 1997 to December 1999, the so-called "gap years" ? Are these measurements ignored? Or, are they fit somehow in a manner that allows for better calculation of other regressors, perhaps with a flat line for the ODS term? b) is the end of the regression line for the first period of time forced to match the start of the regression line for the second period of time?

I think based on the nomenclature of equations (1) and (2) that data in the "gap years" are actually used to help estimate the other beta terms. A simple written explanation would be quite helpful.

Lines 222 to 223: The phrase "uncertainties were calculated with error propagation", without even a citation, are too vague to be of any use to the reader who may one day attempt to reproduce these results. In my opinion, either more detail needs to be added here (my preference), or else a very specific citation such as "as described in Section X.Y of Author et al." should be added.

Line 226: Connecting to the first major comment above, for Table 1 (line 111), I was surprised to read "the eight merged satellite datasets" because I had remembered reading that Table 1 showed ten datasets.

Lines 243 to 244: this one sentence paragraph contains important information that I cannot fully understand, since the sentence is so short. Is it the 4 additional years to the proxies (i.e., a temporal extension) that has a small impact? Or, perhaps, is it updates to proxies for ENSO, QBO, etc that has a small impact. More detail here, that is, an actual paragraph, would be helpful. Also, please name the proxies.

Line 295: Figure 3b is a very important result. Given the anomalous behavior of ozone in year 2024. I suggest either for main or supplement a figure showing how the trend in stratospheric column ozone, as a function of latitude, varies for different choices of the end year. Although this next point is "minor" and easy to address, I'll include here as this point is a natural follow-on. For lines 304 and 305, where the latitudinal dependence of the ozone trend is compared to WMO-2022 and to Weber et al., 2022, would be great if the time period could be stated.

Lines 333 to 334: Following the same thread as just above, a more detailed explanation about the event that led to "high ozone levels observed in NH in 2024" (this phrasing could be improved) would be very helpful for the reader. Why force an interested reader who might

not be familiar with Newman et al., 2024 to fetch this paper, to obtain a senses of the forcing that led to the high ozone?

Lines 364 to 369: Here, we have a discrepancy in ozone trends over Lauder, NZ found by various observing methods. Again, the text is terse that the reader has to do their own work to find out how often the ozonesondes are launched, whether there has been any attempt to compare ozone abundances reported by the ozonesondes to the other measurements for time periods overlap, etc. More detail here would be quite helpful. For the record, I have thought long and hard about whether to make such requests for more detail. I feel fine making this request, given the large number of co-authors on this paper with enormous expertise, including two colleagues who work at the Lauder station who undoubtedly can take on this task.

Lines 397 to 399: Are the three merged datasets used here the only ones with longitudinal resolution? If so, this detail would be nice to add to the paper. If not, then some reason for the selection of these three would be helpful. Finally, why 2003-2024 here, when prior sections have focused on 2000-2024? Again, a simple explanation would be quite helpful. It seems this question is answered on line 433; if so, this detail should be appear earlier, and can be repeated here.

**Minor comments:**

Lines 41 and 88: suggest "chemistry-climate models" rather than "climate models"

Lines 45 to 46: this sentence does not represent the discrepancy seen over the Lauder station. Suggest a new sentence or two here, that: names the stations (or region of the stations for the 7 used for the "Apline" region), describes the types of measurements, and makes at least passing mention of the one discrepancy that emerges (that is, ozonesondes over Lauder).

Line 142: it is a shame that the models used an ODS scenario from WMO (2018), since the ODS scenario in WMO (2022) is so different. I suggest adding a sentence or two describing these differences, either here or in Summary section. This paper by Lickley et al. https://acp.copernicus.org/articles/24/13081/2024/ would be a good source of information, for these differences.

Line 145: suggest "stratospheric aerosol forcings"

Lines 149 to 150: the sentence "Compared to ... forcing" is quite vague. A sentence or two providing more information on these "changes" would be helpful to the reader. Also, perhaps again "stratospheric aerosol forcing".

Line 164: suggest "fit" rather than "fitted"; terribly minor suggestion and wow, the paper is so well written I even hesitate with this suggestion.

Line 202: I think the text is referring to the first term on the right-hand side of Eq. (4). Otherwise, I am confused. If so, please clarify here as well as when the phrase "second term" is used, a few lines down.

Line 232: Suggest "trends of ..." and "upper stratosphere, that are ..."

Line 277: I find the phrase "reporting slightly stronger positive trends in the SH middle stratosphere" to be confusing. Stronger in the middle stratosphere compared to the lower stratosphere? Or, stronger within the "several merged datasets" that is the subject of this phrase, compared to other datasets?

Line 347: if the word "assessment" is maintained, then an assessment should also be cited. Otherwise, can go with "evaluation" since GB22 is not an assessment.

Line 349: suggest adding "over 2000 to 2024" somewhere at the start of this paragraph, even though this detail is also in the caption of Figure 6.

Line 364: here and other places where appropriate, I would write "-2.5 to -6 % decade^-1". ON line 373, we see "reaching +2.5-4 % decade^-1" I would write this as "reaching +2.5 to +

4 % decade^-1", or perhaps "reaching 2.5 to 4 % decade^-1".  My point is sometimes "-" is used to mean "to", and other times "-" is used as a minus sign,

Lines 370-386: Again, the "2000 to 2024" detail should be interspersed in a place or two

Line 388: I can not distinguish the color used for Lidar from the color used for satellite mean. Also, "gray shading" should be added to the caption.

Line 419: text says "seen especially in SCIAMACHY-OMPS" but to me, this small negative trend appears to be seen only in SCIAMACHY-OMPS.  Mighty be a hint of a negative trend in one other dataset.